# The Effect of Insect Bite Hypersensitivity on Movement Activity and Behaviour of the Horse

**DOI:** 10.3390/ani13081283

**Published:** 2023-04-08

**Authors:** Denise Söderroos, Rickard Ignell, Pia Haubro Andersen, Kerstin Bergvall, Miia Riihimäki

**Affiliations:** 1Department of Anatomy, Physiology and Biochemistry, Swedish University of Agricultural Sciences, 750 07 Uppsala, Sweden; 2Department of Plant Protection Biology, Swedish University of Agricultural Sciences, 234 22 Lomma, Sweden; 3Department of Clinical Sciences, Swedish University of Agricultural Sciences, 750 07 Uppsala, Sweden

**Keywords:** insect bite hypersensitivity, *Culicoides*, equine, allergy, dermatology

## Abstract

**Simple Summary:**

Insect bite hypersensitivity (IBH, “sweet itch”) associated with *Culicoides* biting midges is the most common allergic skin disease in horses, seriously reducing the welfare of affected horses. This study (1) investigated the effect of IBH on animal welfare and behaviour and (2) assessed a new prophylactic repellent. There were no differences in movement activity and observed behaviour between IBH-affected horses and control horses. However, horses displayed more itching behaviours (e.g., body shaking and scratching) in the evening than in the morning and should therefore be stabled/protected by, e.g., insect repellents and a protective horse blanket in the evening, when *Culicoides* are most active. Even short periods of scratching were associated with moderate/severe inflammatory skin lesions. In order to improve welfare in IBH-affected horses, even short-term exposure to *Culicoides* should be avoided. Preliminary results indicated that the new repellent can be used as a safe, non-toxic, environment-friendly prophylactic to potentially reduce allergen exposure and prevent signs of IBH, although further studies are needed to determine its efficacy.

**Abstract:**

Insect bite hypersensitivity (IBH) associated with *Culicoides* biting midges is a common allergic skin disease in horses, reducing the welfare of affected horses. This study investigated the effect of IBH on animal welfare and behaviour and assessed a new prophylactic insect repellent. In total, 30 horses were recruited for a prospective cross-over and case–control study. Clinical signs of IBH, inflammatory markers in skin biopsies and behavioural data (direct observations, motion index) were scored longitudinally during two consecutive summers. No differences were observed in the total number of itching behaviours or motion index between IBH-affected horses and controls, but higher numbers of itching behaviours were observed in the evening. IBH-affected horses showed both clinical and histopathological signs of inflammatory skin lesions, with even short periods of scratching being associated with moderate/severe inflammatory skin lesions. In order to improve the welfare of the IBH-affected horses, they should be stabled/given extra protection in the evening and even short-term exposure to *Culicoides* should be avoided. Preliminary results showed that the repellent tested can be used as a safe and non-toxic prophylactic to potentially reduce allergen exposure in horses with IBH, but further studies are needed to determine its efficacy.

## 1. Introduction

Insect bite hypersensitivity (IBH) or summer eczema (“sweet itch”) is the most common chronic allergic skin disease in horses worldwide and has a severe negative effect on the welfare of affected horses. The disease can affect all breeds, irrespective of age and sex [1,2]. The prevalence of IBH varies widely (3–60%) between countries and between breeds, but Icelandic horses imported from Iceland to the European continent are particularly affected (>50%) [3]. This can probably be explained by the absence of *Culicoides* biting midges in Iceland and thus lack of exposure of horses to the antigens from the biting midges at a young age [4]. In Sweden, the reported prevalence in such horses is 26–35% [1,4,5,6]. The affected horses react mainly against antigens present in the saliva of *Culicoides* biting midges and occasionally antigens from other biting insects, e.g., *Simulium* black flies [1,7,8,9,10,11]. The main clinical sign at onset is severe pruritus caused by hypersensitivity reactions to insect bites during the warmer months (spring–autumn), which is the active season for biting insects [1,7,8,9]. The clinical signs typically regress during winter, in the absence of exposure, but in severe chronic cases and in warmer climates, they may persist during winter/year-round.

The preferred feeding sites of the *Culicoides* are at the base of the mane, base of the tail, ear pinnae, intermandibular area and ventral midline [5,12], which are also the most prominent lesional areas. Crusted papules, lichenification and dermal oedema with skin folding and self-trauma (scratching) induced fractured hairs (“rat tail”, “buzzed off mane”), excoriations with open wounds and crusting are typical clinical signs. In addition, pigmentation disturbances and secondary bacterial infections can occur [13,14]. Horses may also show behavioural changes, such as increased restlessness, depression, anxiety, nervousness and, due to these, loss of weight [15].

While the suffering and pain related to an open wound are rather obvious, less is known about how itching contributes to impaired welfare in horses. In humans, itching (pruritus) can cause enormous suffering [16] and reduce the quality of life. The itch signal is initiated by a complex interaction between skin cells and nerve fibres and is transmitted to the central nervous system by C-fibres, which also convey pain. The scratching behaviour in response to pruritus is highly relieving and rewarding [16,17], but also contributes to the further release of pro-inflammatory mediators, driving the condition. The pain induced by scratching the skin suppresses the itch temporarily, but this behaviour may lead to decreased periods of rest and development of stress [18]. Such periods may therefore be associated with increased overall movement activity, but this has not been studied previously in the IBH-affected horse. If movement activity is correlated with pruritus, it could be used as an indicator in monitoring insect attacks or, e.g., assessing the effect of treatment for IBH.

The current clinically recommended treatment for IBH is to avoid exposure to insect allergens, especially those of *Culicoides*. The most common method to achieve this and prevent disease in IBH-affected horses is by physical protection with a full-body blanket [19]. Another recommendation is to keep affected horses in adequately closed stables in an insect-proof environment from mid-afternoon to mid-morning [20]. In production animals, prophylactic insect control protocols involving insecticidal repellents such as pyrethrins (permethrine and cypermethrine) can be partly effective [21]. However, pyrethrins are toxic to aquatic organisms, so a less environmentally problematic approach would be to improve the efficacy of the commercially available mosquito traps [22]. This could be achieved using semiochemicals, which are attractants used by the insect during location and discrimination of a suitable host. Another semiochemicals-based approach could be to use non-host volatiles in repellent collars or placed on a protective blanket to repel blood-feeding insects [23,24,25,26,27]. However, the possible benefit of such a strategy has not yet been tested in horses. The aim of this study was therefore to test a new strategy involving use of an insect repellent to reduce the number of insect bites from *Culicoides* and thus exposure to allergens in IBH-affected horses. Furthermore, the aim was to investigate the effect of IBH on movement activity and behaviour of the horse. The hypothesis was that IBH-affected horses display higher movement activity and perform more itching behaviours than non-IBH-affected control horses.

## 2. Materials and Methods

### 2.1. Study Plan

A prospective cross-over and case–control study was performed during two consecutive summers (2019–2020). The study plan was approved by the Regional Ethical Review Board, Sweden (5.2.18-8707/14). Horses that participated in both 2019 and 2020 were located in the same geographical area in both summers. Each participating stable had one or more IBH-affected horses and one non-affected control for each horse with IBH; although in some cases, the same control was used for several IBH-affected horses, due to the lack of available controls in the same paddock.

At the start of the study, information about the horses (e.g., gender, breed and age) was collected using a questionnaire. The questionnaire also included questions about the main use of the horse, hours per day kept outside, size of the paddock and use of a protective horse blanket. Two different questionnaires were created, one for the controls and one for IBH-affected horses. For the IBH-affected horses, questions about when the onset of IBH was noticed and about the treatments currently used for IBH were included. From June to October 2019, randomly selected horses with IBH were equipped with a collar (Horsepol, Kobyłka, Poland), fitted with a small metal box with a perforated lid on the lateral side. A sachet with a novel formulation of insect-repellent odour, based on a blend of four non-host volatile organic compounds identified in cattle [25,26,27], was placed in the box. Horse owners were requested to replace the sachet with a novel formulation every three weeks. The collar was intended to be kept on the horse at all times, except during exercise. Some practical problems with the repellent collars and boxes were observed during the 2019 season, so in 2020 two sachets of the repellent were placed in small synthetic mesh bags that were sewn onto the protective blanket (one in the neck region and one close to the tail), and all participating IBH-affected horses (*n* = 8) were provided with the novel repellent. All stables participating in the study were provided with a Mosquito Magnet^®^ trap, baited with the commercial 1-octen-3-ol lure (Woodstream Corporation, Lancaster, PA, USA), which was placed outside the paddock in order to attract and capture host-seeking mosquitoes and *Culicoides*. The effect of the novel insect repellent was assessed based on clinical assessments of IBH and skin inflammation markers observed in collected skin biopsies.

### 2.2. Horses

In total, 30 convenience-sampled horses were included in the study, 16 with IBH and 14 controls with no clinical signs of IBH. The IBH-affected horses were recruited based on owners’ reports of clinical signs of the condition in the previous year, despite the use of prophylactic methods. The presence of the condition was confirmed using a clinical scoring system on three occasions during the first season (2019). The controls were selected based on two criterias: (1) they were kept in the same paddock as the corresponding IBH-affected horse and (2) they had no previous or present clinical signs of IBH, based on owner reports and clinical score assessments in 2019. The horses included in the study were of different breeds, ages and sexes. The mean age of IBH-affected horses and controls at the start of the study was 14.6 years and 15.7 years, respectively (Table 1). Both IBH-affected horses and controls were kept outside for eight hours or more per day during the study period. Each IBH-affected horse and its control were kept in the same paddock during the whole study period. Paddocks varied in size, but all were 1 hectare or larger. The majority of the paddocks had trees, buildings or other fixed objects that horses could use for itch-related behaviours. During the experiment, horses were fed, kept and treated as usual and treatmens and a protective horse blanket were used for some horses in 2019. In 2020, blankets were used on all IBH-affected horses. The majority of the IBH-affected horses also received an additional treatment, such as emollient cream or cortisone cream, to reduce pruritus, chlorhexidine for treatment of infections or emollient shampoo.

### 2.3. Clinical Assessment and Biopsy Collection

IBH-affected horses were clinically examined and the severity of the IBH was graded on a total of three occasions during each summer using a clinical scoring system to grade five types of skin lesions (alopecia, excoriation, lichenification, crust and oedema) and four skin thickness categories (0 < 5 mm, 5–10 mm, 10–15 mm and >15 mm) on a scale from 0 to 3 (0 = normal, 1 = mild clinical signs, 2 = moderate clinical signs and 3 = severe clinical signs) for each of 10 body areas (head, ears, neck, mane, back, tail, flank, croup, ventral midline and legs). The maximum total lesion score possible was 120. Control horses were also examined to ensure that they had no clinical signs of IBH. In 2019, clinical signs of IBH were assessed in 15 IBH-affected horses and 13 control horses (Table 2). Nine IBH-affected horses and eight controls were assessed in both 2019 and 2020 (Table 2). In early and late summer 2019 and 2020, two to three skin biopsies were taken for each IBH-affected horse and analysed for in total eight horses that participated in the study in both years (Table 2). The biopsies were collected from areas with typical skin lesions indicating IBH, or from the base of the mane or tail when the skin appeared normal at clinical examination. After collection, biopsies were placed in Histofix^®^ solution (1–2 samples per horse) and RNAlater^®^ RNA stabilisation solution (1 sample per horse). One day after biopsy collection, samples placed in RNAlater^®^ were stored at −80 °C until analysis and samples placed in Histofix^®^ were stored at −20 °C. Haematoxylin–eosin (HE) and periodic acid-Schiff (PAS) stained biopsies were subjectively graded for level of inflammation by a blinded veterinary pathologist. Epidermal hyperplasia, signs of dermal inflammation, presence of eosinophil granulocytes and crusts, erosions and spongiosis were each graded on a scale from 0 to 4, where 0 = no signs of inflammation, 1 = mild signs of inflammation, 2 = moderate signs of inflammation and 3 = severe signs of inflammation.

### 2.4. Movement Activity

IceTag^®^ accelerometers (IceRobotics Ltd., Edinburgh, UK) were mounted on the lateral side of a leg pad and placed on a randomly selected hind leg of the horse. These accelerometers measure motion index, which is a proprietary measure of the overall activity of the animal measured in three dimensions, number of steps and lying time. The accelerometers were used to measure movement activity continuously for approximately 7 days, in the period May–August 2019 and 2020 for control horses and horses with IBH. All matched pairs (one IBH-affected horse and one control) were measured at the same time. In 2019, movement activity was measured in a total of 20 horses (11 IBH-affected horses and nine controls, Table 2). The IceTags^®^ were unintentionally detached from some horses, but were re-attached as soon as this was noticed. For three of these horses (one IBH-affected horse and two controls), the IceTags^®^ could not be found, and therefore, movement data from these horses could not be analysed. In 2020, movement activity was measured in a total of 11 horses (six IBH-affected horses and five controls, Table 2), but for one of the IBH-affected horses, the number of steps was not measured due to technical failure of the IceTag^®^. Therefore, data from that horse and its control were not included in the statistical analysis. In total, five IBH-affected horses and four controls were included in the study in both years (Table 2).

The accelerometers were removed from the horses during exercise and when the horses were taken outside from the pasture, and the time and duration were noted. Data from periods when accelerometers were removed or unintentionally detached were excluded from further analysis. After the measurements, the IceTags^®^ were read in an IceReader^®^ (IceRobotics Ltd., Edinburgh, UK) and processed in the IceManager^®^ 2014 software (IceRobotics Ltd., Edinburgh, UK). The data were exported to Microsoft^®^ Excel 2016 (Microsoft, Redmond, WA, USA) and non-relevant time periods, e.g., during exercise, were removed. When a period was removed for one horse, the same period was removed for the other horse in the pair.

### 2.5. Direct Observations of Behaviours

Direct observations using an ethogram of insect-repellent behaviours (Table 3) (modified from Hartmann et al. [28]) were performed on 11 horses; six IBH-affected horses with average age 14 (range 2–26) years, and five controls with average age 16 (range 7–24) years. These horses, kept at five different stables, were studied between June and August in the two consecutive summers (Table 2). In summer 2019, direct observations were also made for another six IBH-affected horses and five controls, in addition to the 11 horses observed in both 2019 and 2020 (Table 2). The horses (*n* = 22) observed during 2019 were located at seven different stables (Table 2). Observations were performed for 60 min and documented using video recordings, on two occasions per horse, during the period when IceTags^®^ were measuring movement activity. Control horses were observed during the same period as the IBH-affected horses, with in total two to four horses observed at the same time. One observation was performed in the evening (start of observation between 18.30 and 21.30 h) and one in the early morning (start of observation between 05.15 and 07.30 h), when horses were in the paddocks. Horses that were not outside in the mornings were observed twice in the evening instead. The observer was located either outside or inside the paddock at a distance from the observed horses, from where it was possible to see the behaviours but minimise the interaction and the risk of affecting horse behaviour. Behaviours performed by the horses were noted in a protocol and later exported to Microsoft^®^ Excel 2016 (Microsoft, Redmond, WA, USA).

### 2.6. Data Analysis and Statistics

All statistical analyses were performed in SAS software (version 9.4; SAS Institute Inc., Cary, NC, USA). Due to the loss of horses in the second year, separate analyses for itching behaviour and clinical score for IBH severity were performed for IBH-affected horses and controls for 2019 and for horses that participated in the study in both 2019 and 2020. For analysis of movement activity, data were analysed separately for each year and for horses that participated in both 2019 and 2020. Number of steps and lying time data from horses participating in both 2019 and 2020 as well as data from direct observations and clinical assessments were square-root-transformed before statistical analyses in order to obtain a normal distribution of the residuals. The effect of IBH on the number of observed itching behaviours during the 60 min observation period was analysed with a general linear mixed model (Proc Mixed). Stable, group (IBH-affected or control), observation, time of day (morning or evening), weather (sunny, cloudy or rainy) and use or not of a protective horse blanket were included in the statistical model as fixed effects and horse as a random effect.

The effect of IBH on the clinical score was also analysed using a general linear mixed model (Proc Mixed) with stable, group (IBH-affected or control), time (early summer, summer or autumn) and use or not of a protective horse blanket as fixed effects and horse as a random effect. When analysing the effect of season (early summer, summer or autumn) on scores given to IBH-affected horses, group was removed from the statistical model. Comparisons of skin inflammation markers in biopsies taken in 2019 and 2020, and between biopsies taken during early summer and autumn, were performed using a general mixed model (Proc Mixed). The model included time (early summer or autumn and 2019 or 2020), use or not of a protective horse blanket and use or not of treatment for IBH as fixed effects and horse as a random effect. A mixed model (Proc Mixed), including season (early summer or autumn), use or not of a protective horse blanket and use or not of treatment for IBH as fixed effects and horse as a random effect, was used when analysing clinical scores and skin inflammation markers in biopsies. A general linear model (GLM) was used to investigate the effect of IBH on movement activity for horses participating in 2019, while a general linear mixed model (Proc Mixed) was used for horses participating in both 2019 and 2020. The GLM included group (IBH-affected horse or control), stable and use or not of a protective horse blanket as fixed effects (except when analysing data from 2020) and age as a continuous effect. An average per minute was calculated for movement activity and the number of steps. Values from direct observations and clinical assessments are presented as median and range and those from IceTags^®^ and biopsy assessments as least square mean (LSM) ± standard error (SE), unless otherwise stated. The significance level was set to *p* < 0.05, with a tendency for significance at *p* < 0.1.

## 3. Results

### 3.1. Horses Included in the Study

Horses were included in the study based on clinical signs and the likelihood of remaining at the same stables for the whole study period. However, eight IBH-affected horses and seven controls were withdrawn during the study period (Table 2). For the IBH-affected horses, the main reason for this was euthanasia due to a diagnosis other than IBH (such as laminitis) or because the horse was sold off during the study period. For controls, the main reason for leaving the study was the withdrawal of the paired IBH-affected horse. No new horses were recruited for 2020. In 2019, 12 IBH horses carried a collar with the novel insect repellent, while eight IBH-affected horses were aimed to be used as controls. However, seven (of the eight) IBH-affected horses that did not carry the collar during summer 2019 were sold, euthanised or left the study before the 2020 season. Four of the IBH-affected horses that carried the insect-repellent collar during summer 2019 were sold (*n* = 3) or euthanised before spring 2020. Due to missing data (horses excluded from the study) and technical problems with collars in 2019, the IBH-affected horses that carried an insect-repellent collar in 2019 could not be used as their own controls during summer 2020.

### 3.2. Itching Behaviours

There was no difference in the number of total itching behaviours between IBH-affected horses and controls (*p* > 0.5). An effect of stable and time of the day was found, with a higher number (*p* < 0.05) of itching behaviours observed in the evening than in the morning (median 10, range 0–43 vs. median 1, range 0–24, respectively). Year, weather and use of an insect-protective horse blanket did not affect the number of itching behaviours performed by the horses (*p* > 0.05). During the direct observations in 2019, horses performed a higher number (*p* < 0.05) of itching behaviours during sunny weather (median 8, range 6–36) compared with cloudy weather (median 2, range 0–9).

### 3.3. Movement Activity

Movement activity was measured in nine IBH-affected horses and eight controls in 2019 and in five IBH-affected horses and four controls in 2020. Mean motion index per minute, mean number of steps per minute and mean lying time (min) did not differ between IBH-affected horses and controls during 2019 or 2020 (Table 4a, *p* > 0.05). Stable had an effect on the number of steps per minute in 2019 and on the motion index per minute in 2020 (Table 4a, *p* < 0.05). In 2019, lying time (min) tended to be affected by stable (Table 4a, *p* < 0.1). When analysing data for horses included in both years (five IBH-affected horses and four controls), a tendency for higher average motion index per minute was found for IBH-affected horses compared with controls and for horses that used a protective horse blanket (Table 4b, *p* < 0.1). Motion index per minute was affected by stable (Table 4b, *p* < 0.05). Mean number of steps per minute and mean lying time did not differ between IBH-affected horses and controls (Table 4b, *p* > 0.05). Number of steps per minute was affected by stable (Table 4b, *p* < 0.05). Horses that were studied in both years had a daily mean ± standard deviation (SD) of 126 ± 46 min.

### 3.4. Clinical Signs of Allergic Dermatitis

In 2019, IBH-affected horses had a higher overall score for clinical lesions associated with allergic dermatitis compared with the controls (median 6, range 1–35 vs. median 0, range 0–5, respectively, *p* < 0.05). During the same year, IBH-affected horses received lower lesion scores in early summer than in summer and autumn (*p* < 0.05), but there was no difference between the scores for summer and autumn (*p* > 0.05) (Table 5a). Stable and use of an insect-protective horse blanket did not affect the scores given to IBH-affected horses in 2019 (*p* > 0.05). IBH-affected horses received higher lesion scores in autumn 2019 compared with early summer and summer 2020 (Table 5b, Figure 1, *p* < 0.05). The same horses were given higher lesion scores in summer 2019 compared with early summer 2020 (Table 5b, Figure 1, *p* < 0.05). When including both years in the statistical model, lesion scores were affected by stable (*p* < 0.05), but not by the use of an insect-protective blanket (*p* > 0.05). Overall, horses received higher lesion scores in 2019 than in 2020 (*p* < 0.05).

### 3.5. Skin Biopsies

Skin biopsies from eight IBH-affected horses were examined. Two of the horses were assessed as having skin inflammation in autumn 2019, while none had skin inflammation in early summer 2019 and early summer and autumn 2020. Total scores in the biopsy assessment were higher in autumn compared with early summer (Figure 2, *p* < 0.05). Overall, no differences were found between scores in 2019 and 2020 (Figure 2, *p* > 0.05). Horses that had a protective horse blanket received higher scores (3 ± 0.5) than horses without a blanket (1.5 ± 0.5) (*p* < 0.05). No effect of medication (e.g., local or systemic anti-inflammatory drugs) on the total score was found (*p* > 0.05).

## 4. Discussion

In this study, IBH was not associated with increased movement activity or the number of itching behaviours. However, IBH-affected horses showed clinical lesions associated with scratching (e.g., alopecia and excoriations), so the itching behaviours leading to these injuries must have been performed during periods of the day when they were not observed. One possible explanation is that the two direct observation periods per horse (60 min each) were too short to detect itching behaviours. This is interesting, since the observations were performed in periods (morning and evening) when *Culicoides* are most active [29]. Horses displayed more itching behaviour in the evenings than in the mornings, which agrees with findings in entomological studies that *Culicoides* are most active in the hours just after sunset [29,30]. This highlights the importance of adequate protection from *Culicoides* for affected horses during that period. There was also great variation in the availability of physical objects to scratch against at the different stables. In many cases, the most severely affected horses had limited access to trees or other objects in their paddock, which may have limited their expression of itching behaviour. In addition, the majority of the horses were stabled for short periods during the day, e.g., before riding or for grooming, and in the stable, they had the opportunity to scratch against box walls, etc. Field observations showed that horses without IBH also performed itching behaviours, but these horses received low lesion scores and had no alopecia or excoriations. The horses may have performed other itching-related (insect-repelling) behaviours, such as rolling, body shaking and lifting the hind leg, because of irritation by insects rather than due to itching. In a previous study, exposure to insects was found to have an effect on both insect-repelling behaviour and saliva cortisol levels in non-IBH-affected horses, indicating a negative effect on the welfare of just the presence of insects [31].

Stables in different parts of Sweden and in different local biotopes were included in this study, as previous studies have demonstrated an association between IBH prevalence/severity and geographical location or particular local biotopes [5,20,32]. In order to objectively quantify the seasonal and local variation in *Culicoides*, Mosquito Magnet^®^ traps were used for the collection of *Culicoides* [33], and owners were instructed to save all collected insects. Unfortunately, several horse owners encountered practical problems in performing this task, and due to missing data, information on seasonal and local variation in *Culicoides* is not presented in the results. Anecdotally, there are several owner-based observations of improved welfare in IBH horses when using these traps, but scientific studies on effective prevention of IBH signs are still lacking.

In the clinical assessments, more severe signs of IBH were recorded in 2019 than in 2020, when comparing the same horses in both years. This may indicate a greater incidence of *Culicoides* in 2019, but, as mentioned, the incidence could not be measured. On comparing skin inflammation markers from biopsies, no differences between the years were found, so the difference between the years in scores from the clinical assessments could have been due to discrepancies in scoring by the two assessors, rather than to the number of *Culicoides* present in the field. In previous studies, different standardised protocols have been used by multiple veterinarians performing assessments of dermatitis in IBH-affected horses, and a recent study showed that different clinical scoring systems have an excellent ability to correctly determine the severity of IBH in horses [19,34]. Based on this, it is likely that the detailed protocol used in this study gave similar inter-observer results.

During summer 2020, all IBH-affected horses were supplied a novel insect repellent [26,27], which was placed in mesh bags close to the mane and tail. There were no owner-reported changes in social interaction or any adverse skin reactions in horses that were equipped with the novel insect repellent. These preliminary findings indicate that semiochemicals can be used as a safe, non-toxic, environment-friendly prophylactic strategy in IBH-affected horses. However, confounding factors can have affected the results, such as the possibility of seasonal/yearly variation in the amount of *Culicoides*, and the potential repellent effect of the semiochemicals on the healthy control horses kept in the same paddock as the treated IBH-affected horses. Further studies are needed to evaluate the efficacy of the semiochemicals in reducing allergen exposure and preventing clinical signs of IBH. For ethical reasons, other conventional prophylactic measures (e.g., horse blanket and insect repellent) were not withheld from the IBH-affected horses during the experimental part of the study. Despite this, horses receiving these prophylactic treatments still showed clinical signs of IBH, indicating that they were exposed to *Culicoides.* Horses that wore a protective horse blanket had more markers for IBH according to the biopsy assessment, which was unexpected since a blanket provides physical protection from biting insects. However, it is possible that horses showing less severe signs of IBH, and therefore receiving lower lesion scores, were not given a protective blanket.

Apart from welfare concerns, the commercial value of IBH-affected horses may be reduced. Seven of eight IBH-affected horses that were not equipped with the insect repellent in 2019 were excluded from the study for different reasons (sold (*n* = 1), euthanised (*n* = 2), other reasons (*n* = 4)). Before spring 2020, an additional four IBH-affected horses were withdrawn from the study (sold (*n* = 3), euthanised (*n* = 1)), of which three were not equipped with the insect repellent in 2019. Only one of the control horses was sold during the study period and none of the controls was euthanised. Based on this, there appears to be a higher risk of horse owners choosing to euthanise or sell an IBH-affected horse. This reflects the fact that choice of treatment and prevention of this distressing disease remain major challenges for the veterinary profession and for horse owners, as there is still little evidence-based data on disease management. Understanding of the disease at genetic and immunological levels has improved in recent years, but the welfare of affected animals would benefit greatly from science-based guidelines on prophylaxis and treatment [15]. New disease outbreaks caused by vector-borne infectious diseases in horses are an increasing threat to animal welfare and the horse industry [35,36].

The main weakness of this study was the large number of horses that left the study before the second season, which resulted in a small sample size. However, no new horses were recruited for season two, and data from the same horses were included in the statistical analysis when comparing 2019 with 2020. Another major problem was the faulty attachment of the boxes containing the novel insect repellent during the first year, which resulted in only a small number of horses being treated with the repellent. As a result, no conclusions could be reached about the efficacy of insect repellent in 2019. However, in 2020, all IBH-affected horses were treated with the novel insect repellent throughout the study period, and some of the results indicated a potential positive effect of the product (lower clinical scores in 2020). Horses of different breeds were included in the study (Table 1), and in many cases, the IBH-affected horse and its paired control were not of the same breed, which may have influenced the results. For example, Icelandic horses and New Forest ponies seem to develop more severe clinical signs of IBH compared with Finnhorses (2). Movement activity may also be affected by breed. An important criterion was to have the IBH-affected horses and their controls in the same paddock, in order to minimise the risk of these groups being exposed to different amounts of *Culicoides* and to exclude any possible effect of the paddock on movement activity and itching behaviours. Most of the IBH-affected horses were given some treatment, which may have decreased their signs of IBH and thus affected the results, but this was necessary for ethical reasons. According to the biopsy assessment, only two (of eight) horses were assessed as having skin inflammation. Future studies should focus on horses with more severe signs of IBH.

## 5. Conclusions

In this study, there were no differences in movement activity and observed behaviour between IBH-affected horses and controls. However, horses showed more itching behaviours in the evening than in the morning and should therefore be stabled/protected by, e.g., insect repellents and a protective horse blanket in the evening, when *Culicoides* are most active. It was found that even short periods of scratching were associated with moderate/severe inflammatory skin lesions. Therefore, to improve the welfare of IBH-affected horses, even short-term exposure to *Culicoides* should be avoided. Promising preliminary results obtained in the study showed that non-host semiochemicals can be used as a safe, non-toxic, environment-friendly insect repellent to potentially reduce allergen exposure and prevent signs of IBH. However, further studies are needed to determine the efficacy of the semiochemicals in reducing allergen exposure and preventing clinical signs of IBH.

## Figures and Tables

**Figure 1 animals-13-01283-f001:**
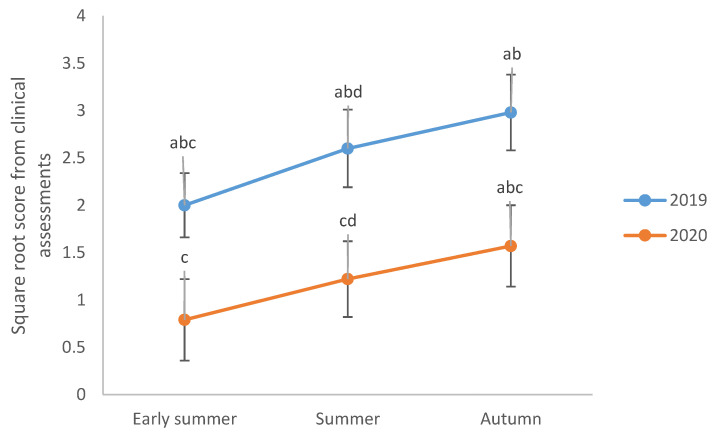
Scores in clinical assessments of the severity of clinical signs in nine horses affected by insect bite hypersensitivity (IBH) in early summer, summer and autumn 2019 and 2020. Values shown (square-root-transformed) are least square mean ± standard error. Different letters (a–d) indicate significant differences between seasons. Overall, horses received higher scores in 2019 compared with 2020 (*p* < 0.05).

**Figure 2 animals-13-01283-f002:**
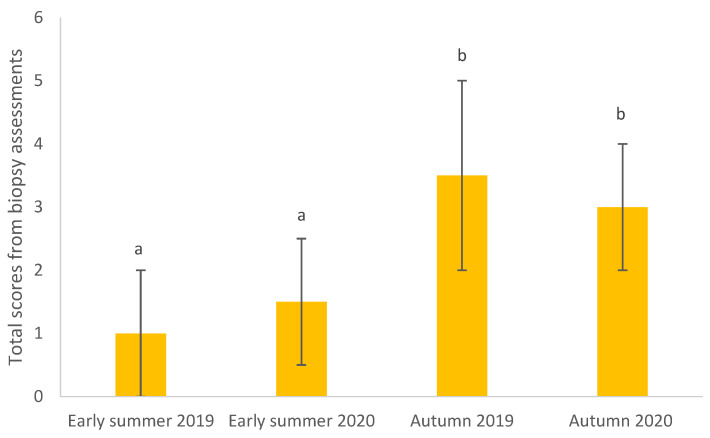
Total scores (least square mean ± standard error) of skin inflammation markers in skin biopsies collected from eight horses with insect bite hypersensitivity (IBH) in 2019 and 2020. Different letters (a, b) indicate significant differences between seasons (*p* < 0.05).

**Table 1 animals-13-01283-t001:** Information on age, breed and gender of horses with insect bite hypersensitivity (IBH) and control horses that participated in the study during at least one season.

Horse	Stable	IBH/Control	Age (Years)	Breed	Gender
A ^1^	1	IBH	17	KWPN	Mare
A ^2^	1	Control	15	Friesian	Gelding
B ^1^	2	IBH	13	Irish Cob	Gelding
B ^2^	2	Control	18	Knabstrup	Gelding
C ^1^	3	IBH	13	New Forest	Mare
C ^2^	3	Control	9	Warmblood	Mare
D ^1^	3	IBH	15	Shetland pony	Mare
D ^2^	3	Control	26	Arabian	Gelding
E ^1^	4	IBH	24	Shetland pony	Mare
E ^2^	4	Control	21	Welsh Mountain	Gelding
F ^1^	4	IBH	19	Dutch pony	Gelding
F ^2^	4	Control	17	Shetland pony	Gelding
G1 ^1^	5	IBH	13	Mixed breed	Gelding
G2 ^1^	5	IBH	26	Dales pony	Mare
G3 ^1^	5	IBH	2	Dales pony	Mare
G ^2^	5	Control	24	Coldblood trotter	Mare
H ^1^	6	IBH	13	Mixed breed	Gelding
H ^2^	6	Control	15	Mixed breed	Mare
I ^1^	6	IBH	12	Connemara	Mare
I ^2^	6	Control	13	Mixed breed	Mare
J ^1^	6	IBH	16	Mixed breed	Mare
J ^2^	6	Control	10	Mixed breed	Mare
K ^1^	7	IBH	5	Polish pony	Gelding
K ^2^	7	Control	8	Mixed breed	Gelding
L ^1^	7	IBH	15	KWPN	Mare
L ^2^	7	Control	11	Warmblood	Mare
M ^1^	8	IBH	9	Icelandic horse	Mare
M ^2^	8	Control	7	Icelandic horse	Mare
N ^1^	8	IBH	22	Icelandic horse	Mare
N ^2^	8	Control	27	Icelandic horse	Mare

^1^ IBH-affected horses. ^2^ Control horses.

**Table 2 animals-13-01283-t002:** Type of data collected for horses affected by insect bite hypersensitivity (IBH) and control horses during 2019 and 2020. A cross indicates that the data type was collected for a horse.

Horse	Clinical Score * 2019	Clinical Score * 2020	Skin Biopsy 2019	Skin Biopsy 2020	Movement Activity and Behaviour 2019	Movement Activity and Behaviour 2020
A ^1^	x ^a^	x	x	x	x	x
A ^2^	x	x			x	x
B ^1^	x ^a^	x	x	x	x	x ^b^
B ^2^		x			x	x ^b^
C ^1^	x	x	x	x	x ^c^	x ^c^
C ^2^	x	x			x ^c^	x ^c^
D ^1^	x	x	x	x	x ^c^	x ^c^
D ^2^	x				x ^c^	x ^c^
E ^1^	x	x	x	x	x	x
E ^2^	x ^a^	x			x	x
F ^1^	x ^a^	Sold	x	Sold	x	Sold
F ^2^	x ^a^	x			x	Not included
G 1 ^1^	x ^a^	x	x	x	x	x
G 2 ^1^	x ^a^	Euthanised	x	Euthanised	Euthanised	Euthanised
G3 ^1^	x ^a^	x	x	x ^a^	x	x
G ^2^	x ^a^	x			x	x
H ^1^	x ^a^	x ^a^	x	Not included	x ^c^	Not included
H ^2^	x	Sold			x ^d^	Sold
I ^1^	x	Sold	x	Sold	x ^d^	Sold
I ^2^	x ^a^	x ^a^			x ^d^	Not included
J ^1^	x ^a^	Sold	x	Sold	x ^d^	Sold
J ^2^	x ^a^	x ^a^			x ^d^	Not included
K ^1^	x	Sold	x	Sold	x	Sold
K ^2^	x	Not included			x	Not included
L ^1^	x	Euthanised	x	Euthanised	x	Euthanised
L ^2^	x	Not included			x	Not included
M ^1^	x	x	x	x	x	x
M ^2^	x	x			x	x
N ^1^	x ^a^	Euthanised	x ^a^	Euthanised	Euthanised	Euthanised
N ^2^	x ^a^	Not included			Not included	Not included

* Clinical scoring of IBH severity. ^1^ Horses with IBH. ^2^ Control horses. ^a^ One or two assessments lacking. ^b^ Movement activity not included in the statistical analysis due to technical failure of IceTags^®^. ^c^ Data not analysed since the IBH-affected horse and the control were not kept in the same paddock during the whole study period. ^d^ IceTags^®^ detached from two of the horses after one day and could not be found, so data from that stable were not included in the statistical analysis.

**Table 3 animals-13-01283-t003:** Ethogram used in behaviour observations.

Behaviour	Description
Rolling	Lying down and moving body side to side
Body shaking	Movements in the whole body at the same time, e.g., when shaking off insects
Scratching with teeth (grooming behaviour)	Scratching with teeth at any body part
Scratching with head (grooming behaviour)	Scratching with the head (often the side of the face) at any body part, usually on the front legs
Scratching with hind leg (grooming behaviour)	Horse brings one hind leg to its head and scratches its head or neck with the hoof [28]
Lifting hind leg	Hind leg moves forcefully up and down, e.g., when shaking off insects or due to irritation of the leg pad
Biting on any body part	Horse bites lightly with the teeth on any body part
Scratching against an object	Horse scratches any body part against an object, e.g., a tree or a building

**Table 4 animals-13-01283-t004:** Average movement activity (least square mean (LSM) ± standard error (SE)) measured by IceTag accelerometers (IceRobotics Ltd., Edinburgh, UK) over approximately 7 days (a) during summer 2019 (nine insect bite hypersensitivity (IBH)-affected horses and eight controls) and summer 2020 (five IBH-affected horses and four controls) and (b) for horses participating in both 2019 and 2020 (five IBH-affected horses and four controls).

(a)								
	**Year**	**IBH-Affected Horses**	**Controls**	**Effect of Group**	**Effect of Stable**	**Effect of Age**		
Motion index/min	2019	17.18 ± 2.37	19.31 ± 3.70	N.S	N.S	N.S		
Steps (n/min)	2019	3 ± 0.4	3 ± 0.6	N.S	<0.05	N.S		
Lying (min)	2019	506 ± 99	638 ± 168	N.S	<0.1	N.S		
Motion index/min	2020	16.75 ± 1.30	15.10 ± 1.40	N.S	<0.05	N.S		
Steps (n/min)	2020	3 ± 0.3	3 ± 0.3	N.S	N.S	N.S		
Lying (min) ^a^	2020	769 ± 212	869 ± 173	N.S	N.S	N.S		
(b)								
		**IBH-Affected Horses**	**Controls**	**Effect of Group**	**Effect of Stable**	**Effect of Age**	**Effect of Protective Clothing**	**Effect of Year**
Motion index/min		21.55 ± 2.7	9.34 ± 3.06	*p* < 0.1	*p* < 0.05	N.S	*p* < 0.1	N.S
Steps (n/min) ^b^		3 (0–4)	3 (2–5)	N.S	*p* < 0.05	N.S	N.S	N.S
Lying (min) ^ab^		623 (337–1200)	810 (653–1273)	N.S	N.S	N.S	N.S	N.S

^a^ Average total lying time (min) measured for approximately 7 days. ^b^ Values shown as median and range. Horses that were studied in both years had a mean daily lying time of 126 ± 46 (SD) min.

**Table 5 animals-13-01283-t005:** Scores recorded in clinical assessments (general linear mixed model (Proc Mixed)) of (a) 15 insect bite hypersensitivity (IBH)-affected horses in early summer, summer and autumn 2019 and (b) 9 IBH-affected horses in early summer, summer and autumn in both 2019 and 2020.

(a)		
**Clinical Assessment**	**Score (Mean ± SD)**	**Score (Median and Range)**
Early summer 2019	5 ± 4 ^a^	3 (1–14) ^a^
Summer 2019	10 ± 7 ^b^	7 (1–26) ^b^
Autumn 2019	15 ± 11 ^b^	14 (1–35) ^b^
(b)		
Early summer 2019	4 ± 3 ^abcd^	3 (2–12) ^abcd^
Summer 2019	5 ± 3 ^abc^	6 (1–8) ^abc^
Autumn 2019	11 ± 11 ^ab^	8 (1–29) ^ab^
Early summer 2020	1 ± 1 ^d^	0 (0–3) ^d^
Summer 2020	3 ± 3 ^cd^	2 (0–8) ^cd^
Autumn 2020	3 ± 3 ^abcd^	3 (0–8) ^abcd^

Different superscripts indicate significant differences between seasons (*p* < 0.05) analysed from square-root-transformed data of scores from clinical assessments (not shown). For horses assessed in both 2019 and 2020, a higher total score was given to IBH-affected horses than to controls (*p* < 0.05).

## Data Availability

The datasets used and/or analysed in this study are available from the corresponding author upon request.

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
