# Peer review of "The Effect of Insect Bite Hypersensitivity on Movement Activity and Behaviour of the Horse"

_animals, 2023, doi:10.3390/ani13081283_

Round 1

Reviewer 1 Report (Previous Reviewer 2)

The revision is suitable for publication after editing of English errors

Author Response

Thanks for your comment, please see attached manuscript after English editing/extensive English revision. 

Reviewer 2 Report (Previous Reviewer 1)

.

Author Response

Thanks for your comment, please see attached manuscript after English editing/extensive English revision. 

This manuscript is a resubmission of an earlier submission. The following is a list of the peer review reports and author responses from that submission.

Round 1

Reviewer 1 Report

It was a very nice and interesting article to read; please consider the following edits. Thank you

Line 20-21: “The IBH-affected horses 20 treated with new prophylactic repellent showed less clinical signs of IBH”. What could be the possible explanation for this? Do repellent helps to avoid further biting of already affected horses? Or it treats the affected horses themselves.

Line 25: Please add examples of itching behaviors here to help your reader.

Line 52-53: “Icelandic horses imported from Iceland to the European continent are particularly affected (>50 %)”. What could be the reasons for it that Icelandic horses are more susceptible to IBH? Please add text here.

Line 61-62: “In general, clinical signs are first observed in horses aged 2-4 years and can worsen with age?” Are horses with ages less than 2-4 years not susceptible to IBH?

Line 68-69: Scratching is also a prominent behavioral sign in horses. If you agree, please add to the text.

Line 87-89: I suggest not writing about production animals here because you already have enough specific text for horses.

Line 109-110: If control is missing for one location. I suggest not using data from that site because it affects scientific methods.

Line 112: Please provide the questionnaire as a supplementary file with this manuscript.

Line 133-134: Equine Eczema Dermatitis Extent and Severity (EEDESI) score. Please cite this method of scoring.

Line 157: As all 38 horses were of different breeds. I suspect that control and IBH-affected animals are of different breeds at one location. How can this affect your overall results, and to what extent is it a confounding factor?

Line 166, 168-170: “Protective horse blankets and treatments were used for some horses.” “The majority of the IBH-affected horses received additional treatment, such as emollient cream or cortisone cream, to reduce pruritus, chlorhexidine for treatment of infections or emollient shampoo.” So, all horses were not treated equally? This is a confounding factor. Please justify.

Lines 444-446 and lines 20-21 give different impressions. Is this a treatment or prevention? What is one possible explanation for this? Does repellent help horses that have already been bitten in your study? Or it treats the affected horses.

Line 476- : Please add a paragraph at the end of the discussion indicating the study’s limitations and confounding factors that may affect results.

Author Response

Explanations of revisions

We want to thank for valuable comments! Please see our response on individual questions below:

Nr 1

Line 20-21: We have now removed that sentence from the summary and abstract and replaced it with another sentence (see the tracked changes). We considered that we could not make that conclusion due to loss of horses and technical problems (we don´t have enough data to measure the effect of the repellent). However, the second year when all of the IBH-affected horses were treated with the repellent, lower scores for clinical signs of IBH were recorded. However, we could not see any differences in skin inflammation markers between years (see the discussion when we mention this).

Line 25: Examples of itching behaviours are added

Line 52-53: Added explanation with reference

Line 68-69: Agree, scratching is a very prominent behavioral sign. In the senctence above, scratching was described as a type of self-trauma. Now we have added scratching as a parenthesis after self-trauma to be clearer. The other sentence then describes other behaviours.

Line 87-89: The text about production animals are included to describe the need for a nontoxic and environmental friendly insect repellent.

Line 109-110: It´s not missing a control for that location, but one horse was the control for two IBH-affected horses at the same location. All of these three horses were kept at the same farm and in the same paddock. We consider that it´s better to include data from these horses because the most important is to have a control from the same location. The results are also presented as least square means (instead of means) which adjusts for uneven groups.

Line 112: Questionnaires added as supplements

Line 133-134: The scoring system was introduced for the research group by our European Diplomate in Dermatology. However we could not find a reference. EEDESI is replaced with clinical scoring (of IBH severity).  In earlier studies, different standardized protocols have been used when multiple veterinarians have done assessments of dermatitis in IBH-affected horses, and a recent study showed that different clinical scoring systems had excellent ability to correctly determine the severity of IBH in horses (line 437-439)

Line 157: Discussed under the Limitations section in the end of discussion.

Line 166, 168-170: We don´t consider it to be a confounding factor since the IBH-affected horses is not compared with each other. Instead, we compare 2019 with 2020 using the same horses each year. Moreover, all of the affected horses were still showing clinical signs of IBH despite the use of treatments.

 2.2 Clinical assessment and biopsy collection

Minor changes are made

Lines 444-447: See line 97-99, the insect repellent is used to prevent horses to be bitten by the insects. It´s not used for treatment of already bitten horses.

Line 476: A paragraph with limitations of the study is added in the end of discussion

Reviewer 2 Report

General

There are 2 aims to this study. The first is to assess the effects of insect bite hypersensitivity on the behaviour of horses, and the second to assess the usefulness of insect repellent is used in cattle in preventing IBH in horses.

As this is a case-control study, correct identification of cases (i.e. identifying horses that are affected by IBH) is of paramount importance. A detailed description of the methods by which candidate horses were identified is required, as is a description of the inclusion/exclusion criteria for cases and controls. What diagnostic methods, if any, were used to confirm that the animals included in the study were experiencing clinical signs caused by hypersensitivity reactions to Culicoides?

Introduction.

The introduction provides sufficient information to enable the reader to understand the rationale for the study.

Methods.

A primary methodological concern is the criteria used to include and/or exclude horses from the study as discussed above.

The methods do not describe how any assessment of the effects of the insect repellent will be made. As this was one of the primary aims of the study this is an important oversight. The problem is complicated by the fact that the authors state that there were “there were practical problems with the repellent collars during the first study year”. The nature of these problems needs to be described somewhere in the manuscript, as does their impact on the data collected. Do these problems affect the validity of the data collected? If not, why not?

The authors state that weather was one of the parameters assessed during the visual monitoring of behaviour, and fine weather as sunny, cloudy or rainy. These are very general descriptions that are of limited value given their lack of specificity. Are more detailed descriptions of weather conditions available? The effects of ambient temperature and wind may also be very important variables that affect behaviour.

Collection of blood samples is described but no details are provided on the analyses performed on these samples.

Results.

It is very difficult to understand which samples were analysed from which horses. A better description of the horses for which results were analysed would be very helpful. This could perhaps be included in a tabular or graphic form.

Horses left the study after the first year and that new horses were recruited during the second. This is obviously unavoidable in a study of this nature, but it raises significant questions about the power of the study. The issues with movement monitors exacerbate this problem. Overall one is left with the impression that the study simply analysed whatever data were available without consideration of the effects of missing data on the validity of the study. This may not be the case and the statement may be unfair but the authors need to address this issue and should include a description of the study’s power in the results section.

In the methods section the authors state that “In early and late summer (2019 and 2020) two to three skin biopsies were collected from horses with IBH”.  However, in the results the authors state that “Biopsies from total eight IBH-affected horses were examined”. Does this mean that only some of the biopsy samples were examined or does it indicate that samples were collected from only some of the affected horses? If so, how are these horses selected? The authors go on to say that “Two of the horses were assessed to have a skin inflammation during autumn 2019, as opposed to none during early summer 2019 and early summer and autumn 2020”. This appears to mean that 2 of 8 apparently IBH-affected horses were showing signs of skin inflammation on biopsy. How can this be explained and how does it impact on the selection of horses for the study?

In the methods section the collection of blood samples is described. No information is provided about the blood samples in the results section.

Discussion.

In the discussion it is stated that “The findings of this study indicate promising preliminary results that semiochemicals can be used as a safe, nontoxic prophylactic, environmental prophylactic strategy.” How and why is this statement justified? As the assessment of the effectiveness of the chemicals was a major aim of the study this needs to be explained clearly. However, I am concerned that the experimental design does not allow an assessment of this type to be made.

Author Response

Nr 2

We want to thank for valuable comments! Please see our response on individual questions below:

2.3 Horses: Inclusion criteria for both IBH-affected horses and controls are added. We did not confirm that the affected horses reacted against the genus Culicoides, however, that is the most common insect that these affected horses react against. In this study, the genus of the biting insects was not of relevance since we only investigated the effect of IBH (regardless of what type of IBH).

2.1 Study plan: A sentence about how the insect repellent were aimed to be tested is now described in the end of the paragraph.

Weather description: Since the effect of weather on the presence of biting insects has been studies earlier, it was not the main aim of our study. Therefore, no detailed descriptions of the weather are available.

Blood samples: No blood samples were analysed, the text about these in the methods was not supposed to be there and is now removed.

Results: A table with description of what type of data that was collected for each horse is now added.

3.1 Horses included in the study. Some numbers adjusted in the text

2.2 Clinical assessment and biopsy collection: The method for this and number of biopsies analysed is now clarified in the text. You can also see Table 3. During the first year (2019), biopsies were collected for all of the IBH-affected horses. During 2020, a number of horses were withdrawn from the study and a total of 8 horses remained in the study both years (and their biopsies were analysed). So to clarify, specific horses were not selected for biopsy collection.

Finally the conclusion regarding semiochemicals are changed  in discussion. We agree that further studies are needed in order to evaluate the efficacy of semiochemicals to reduce allergen exposure ( unfortunately we lost many horses from the study due different reasons ). However to our knowledge this is first time semiochemicals are used in horses as insects repellent and  we hope that these first results will initiate further studies.

Reviewer 3 Report

The is very little new data in the paper. There are some negative results (regarding comparison of movement) some that is common knowledge (protecting the animals at dusk). There are no results regarding the most interesting part the use of non-toxic seminochemicals and the authors can not conclude that their use indicate promising results -lines 438-443 are in contradiction to lines 444 -445. 

Author Response

Thanks for the comments. To our knowledge there are lack of studies were clinical signs and behaviour are studied together on IBH horses. Additionally the conclusion regarding semiochemicals are changed  in discussion. We agree that further studies are needed in order to evaluate the efficacy of semiochemicals to reduce allergen exposure ( unfortunately we lost many horses from the study due different reasons ). However to our knowledge this is first time semiochemicals are used in horses as insects repellent and  we hope that these first results will initiate further studies.